# Examining Psychotherapeutic Processes with Depressed Adolescents: A Comparative Study of Two Psychodynamic Therapies

**DOI:** 10.3390/ijerph192416939

**Published:** 2022-12-16

**Authors:** Ana Calderon, Knut Arne Hooper Storeide, Cecilie Elvejord, Helene Amundsen Nissen-Lie, Randi Ulberg, Hanne-Sofie Johnsen Dahl

**Affiliations:** 1Escuela de Psicología, Universidad Gabriela Mistral, Santiago 7500533, Chile; 2Department of Psychology, University of Oslo, Forskningsveien 3, 0370 Oslo, Norway; 3Faculty of Medicine, University of Oslo, 0372 Oslo, Norway; 4Research Unit, Division of Mental Health, Vestfold Hospital Trust, 3103 Tønsberg, Norway; 5Department of Psychiatry, Diakonhjemmet Hospital, Forskningsveien 7, 0370 Oslo, Norway

**Keywords:** adolescent, depression, Adolescent Psychotherapy Q-set (APQ), short-term psychodynamic psychotherapy (STPP)

## Abstract

To understand processes associated with better or poorer psychotherapy outcomes is vital. This study examined and contrasted interaction patterns between one therapist and two depressed 17-year-old girls, Johanna (good outcome) and Sonja (poor outcome), in short-term psychoanalytic therapies selected from an RCT. Outcome data were collected regarding level of inter- and intra-personal functioning and symptoms of depression. Process data were obtained using the Adolescent Psychotherapy Q-Set on all available sessions. Analyses yielded five relational patterns or “interaction structures” in the two therapy processes; Three explained most of the variance in sessions with Johanna (i.e., ‘positive working alliance’, ‘therapist’s active use of psychodynamic techniques’, and ‘a receptive patient’) and two explained more of the variance in sessions with Sonja (i.e., ‘therapist using a more problem-solving and symptom-oriented approach’ and ‘patient displaying limited capacity for mentalization’). The processes in the two cases presented differences related to mentalization, psychological mindedness, and attachment style of the patients. The therapist used different therapeutic approaches, favouring more psychodynamic interventions in the good outcome case and a more problem-solving and symptom-oriented approach with the poor outcome case. In the latter case, the relationship seemed to be more of a struggle.

## 1. Introduction

A recent systematic review and meta-analysis report that 34% of adolescents globally, aged 10–19 years, are at risk of developing clinical depression [1]. Cuijpers and colleagues [2] found that psychotherapies for depression in youth are effective compared to control conditions, but more than 60% of the adolescents receiving therapy do not respond adequately. In a large study examining the differential effect of three psychotherapeutic interventions, no differences in outcome were observed one year after the end of treatment; the adolescents showed an average of 49–52% reduction in depressive symptoms [3]. Hence, more effective treatments and treatment strategies are clearly needed. Because even if psychotherapy research and the accumulated evidence indicate that psychotherapy do work for some, we know less about *how* psychotherapy leads to different patient responses, especially in therapy with children and adolescents [4]. It is, therefore, vital to examine what goes on in the therapy room that may account for differential outcomes of psychotherapy.

In every therapy process the therapist’s thoughts, feelings and behaviours are contingent on the patient that is in the room, causing a dynamic interaction between the therapists’ and patients psychological functioning that will be unique to each therapeutic dyad [5,6,7,8]. One powerful and easy- to-use observational coding system that could help “to determine not which treatment packages work for which patients, but rather which processes of therapeutic change work for which therapeutic dyads interacting in the real world” [4] (p. 261) is the Adolescent Psychotherapy Q-set—APQ [9,10]. This is because the APQ can be used to analyse and describe in a quantitative manner the complex interactions between therapist and patient in psychotherapy sessions of adolescents. When using the APQ, independent observers draw on audio or video recordings from psychotherapy sessions to rank 100 items on their relative importance within a particular session. Each item describing an element from three domains of the therapeutic process: *The therapist’s* actions and interventions; *The adolescent’s* attitudes, behaviours, feelings, and experiences; *The interaction or the dyad’s* nature and atmosphere. Hence, it can be used to conduct the kind of research that Beutler and colleagues [11] and Høglend and colleagues [12] have suggested is needed in psychotherapy research; an integrative and synergic approach where patient, therapist, procedural, and relationship factors are not researched independently. 

Process researchers have begun to use the APQ to examine what occurs in the consulting room with adolescent clients, with only five studies so far. Calderon, Schneider, Target, and Midgley [13] identified what Jones [14] called “interaction structures” in a sample of 70 patients (35 treated with Short Term Psychoanalytic Psychotherapy [STPP] and 35 with Cognitive Behavioural Therapy [CBT]) from the IMPACT study in the UK [3,15]. Interaction structures are patterns of interaction in a therapist-patient dyad. These patterns are usually repeated across sessions—sometimes without the awareness of the participants—and are associated with changes that may lead to alterations in the patient’s psychological makeup [14]. By using Q-cluster analysis of the sessions, the authors found three interaction structures: Strong working relationship between an emotionally involved young person and a therapist who invites the young person to reflect on experiences and develop self-understanding.Strong working relationship between an emotionally engaged and collaborative young person working with a therapist who actively structured the session to provide space for learning.Difficult working relationship between a non-engaged young person and a therapist working hard to make sense of the young person’s experiences, but without making much progress.

Their main conclusions were that when the relationship between therapists and the depressed adolescent was collaborative, the therapy process was influenced by the therapists’ differential techniques; while in a poor working relationship the techniques used by STPP and CBT therapists alike, were less differentiated and seemed less successful in engaging the young person in the process [13].

In another study, Can and Halfon [16] examined interaction structures within a psychodynamic treatment and how these interaction structures might affect outcome. They included a heterogeneous population of 43 young people and conducted a principal component analysis of the 100 APQ items over 123 sessions, which resulted in five interaction structures:Negative Therapeutic AllianceDemanding Patient, Accommodating TherapistEmotionally Distant Resistant PatientInexpressive Patient, Inviting TherapistExploratory Psychodynamic Technique” (EPT).

Results showed one interaction structure (i.e., EPT) that associated with psychodynamic techniques and four interaction structures related to typical alliance characteristics that can emerge during psychotherapy sessions. The relationship between interaction structures and outcome was conducted using multilevel modeling analyses with Bayesian Markov chain Monte Carlo. None of the interaction structures was found to have a direct effect on symptom change. However, a two-way interaction effect between problem levels at baseline and EPT came forth; patients with higher problems at baseline showed poor outcome in EPT, whilst patients with lower problems at baseline showed good outcome in EPT [16]. Can and Halfon [16] did not use Q-analysis, hence the interaction structures they present are less comparable to those found in the other APQ studies using Q-analysis.

A recent study by Fredum and colleagues [17] used the APQ to examine whether there were unique interaction structures in the early sessions of those adolescents who subsequently dropped out. In a sample of 69 adolescents with major depressive disorder receiving STPP as part of the First Experimental Study of Transference Work–in Teenagers in Norway—FEST-IT [18], 21 were identified as dropouts. APQ ratings from an early session were available and analysed from 16 of these dropout cases. Results from the Q-factor analysis revealed three distinct interaction structures:Mutual trust, collaboration, and the exploration of emotions.Resistance and emotional detachment.Mismatch and incongruence in perception and communication.

The first interaction structure showed a similar interaction pattern as number one in the paper by Calderon and colleagues [13]. Pertaining to the young person who seemed emotional detached and unavailable, the second interaction was rather similar to Calderon and colleagues’ number three. Concerning the therapists in the two studies, they all diverged from their treatment method when the young person seemed to be unengaged, but they differed in the way they diverged. In the IMPACT study the therapist worked toward making sense of the adolescents’ experience, asked for more information, and structured the sessions [13]. In the FEST-IT drop out study the therapist focused on their young patients’ symptoms, encouraged them to discuss assumptions behind their experiences, and refrained from taking position in relation to their thoughts and behaviour [17]. Whether these differences in the therapists’ behaviour when working with adolescents that resisted the attempts of the therapist to engage them, had any effect on dropout, we do not know. Nonetheless, comparison between the three interaction structures found by Fredum and colleagues [17] suggested that the reasons why adolescents drop out of therapy vary and are multidimensional in nature. 

The fourth study on APQ was a single-case study from the STPP arm of the IMPACT study with a 16-year-old girl, Leah, who was diagnosed with depression and borderline personality disorder (BPD) before treatment [19]. Leah showed clinically significant change [20] on symptoms of depression; however, whether she still fulfilled criteria for BPD after therapy is not known. In her treatment, five interaction structures were identified [19]:Therapy process is fluent but does not progress, as therapist challenges Leah’s animated discussion of relationships and her fantasies.Therapy process is stuck as therapist probes Leah’s ‘protective shield’.Therapist pushes through Leah’s expressions of painful emotion to challenge her feelings of helplessness regarding relationships.Leah expresses anger over rejection and injustice but cannot reflect on loss, whilst her therapist challenges Leah’s assumptions.Therapist is gentle in collaboratively exploring Leah’s feelings of depression, powerlessness, and her negative self-perception.

This therapy included little attention to explicit work on the relationship in the here and now (transference work) and the therapist employed techniques not usually recognised as typical of a STPP approach. For example, the therapist offered explicit advice, structured the sessions, adopted a problem-solving approach, and challenged ‘absolute beliefs’, interventions more typically seen in CBT. These deviations technique might be understood as responses to the clinical challenges associated with a patient presented BPD in addition to depression. Whether these alternations enhance outcomes or not for patients with similarly grave difficulties, on both relational and symptomatic level, needs further inquiry.

The final APQ-study was a single-case study of the psychotherapeutic process of a 16-year-old girl whose symptoms became worse at mid-treatment, with self-harming and a suicide attempt [21]. She was treated with STPP in the FEST-IT [18]. Six sessions were coded with the APQ to explore in depth the processes related to the mid-treatment crisis. The APQ scoring indicated that even though there was an overall trusting, working relationship in the last session before the suicide attempt, the young patient expressed vulnerability and feelings of rejection, yet the discourse appeared increasingly rote, and the patient was rather lifeless and communicated in a monotone and affectless way. She did not display feelings of irritability, even when the therapist was quite challenging and made definite statements about what was going on in her mind. The therapist was also directly reassuring, stating that ‘everything would be fine’. While the patient stated she did not seek to be separate or autonomous from her parents; rather she explored being dependent and it seemed like she handed the direction over to the therapist, the therapist encouraged independence [22]. Results might imply that the therapist’s agency increased while the patient was losing hers, leaving her more vulnerable for self-harm and suicidal impulses when she was also feeling rejected, helpless, and uncontained. Again, results indicated that when there were difficulties in the relationship, the therapist deviated from the STPP method like the therapist did with Leah, e.g., made definite statements, were directly reassuring, and provided explicit advice.

To our knowledge, the link between psychotherapy process over whole treatment periods, as captured with the APQ, and outcome has not yet been explored. In the present study, we examine interaction structures that may promote or hinder positive development in time-limited psychodynamic psychotherapy for two depressed adolescents. The focus was on describing the strategies and methods used by the therapist in the cases and how the clients seem to respond (as in the classical clinical case study) but compensating for the weaknesses of solely using subjective data in the classical approach by gathering detailed data, tapping both subjective and objective aspects [23]. Examining the psychotherapeutic process involving two patients with similar problems treated by the same therapist presents a unique opportunity to elucidate factors due to the therapist, the patient, the technique, and their interaction [24,25], and to investigate possible links between process and outcome.

The aim of this study was to examine interaction patterns in two psychotherapy processes with adolescent girls suffering from major depression who achieved different outcomes. We asked:What characterized the two girls and what problems did they present before therapy, immediately after, and at one year follow-up?What interaction structures between therapist and patients were present in the psychotherapeutic processes of both patients?How do these interaction structures develop over time?Are there observable differences in patient behaviour, therapist behaviour, and/or relational aspects between the treatments of the two girls?

## 2. Materials and Methods

### 2.1. The First Experimental Study of Transference Work—In Teenagers (FEST-IT)

Data for this study were obtained from FEST-IT [18], which is a randomized clinical component trial with a design aimed at studying the effects of a significant technique within STPP; transference work (the therapist encourages the patient to explore thoughts and feelings about the therapy and the therapist), for adolescents with major depressive disorder. The study was conducted at Vestfold Hospital Trust in cooperation with the Institute of Clinical Medicine at University of Oslo. The 69 adolescents included in the study were between the ages of 16 and 18. Exclusion criteria were psychosis, pervasive developmental disorders, or substance or alcohol addiction. After agreeing to participate, but before randomization, all adolescents were diagnostically interviewed by one of the researchers in the project.

The present study is a comparative case study, conducted within the framework of a randomized controlled trial (i.e., a case within trial comparison), and is a pragmatic case study [23].

#### 2.1.1. Participants

To investigate differences in therapeutic process, we run a strategic search for two girls, treated by the same therapist, who showed divergent outcomes as measured by the Psychodynamic Functioning Scales [26] at the one-year follow-up interview. Only two patients fulfilled these criteria, and they were called Johanna and Sonja. The patients were 17 years old, attended high school, and lived in a city in the eastern part of Norway. After the income interview tentative case formulations including the central dynamics of depression described by Busch and colleagues [27] were outlined. Johanna’s central depression dynamic seemed to be that her thoughts and experiences revolved around guilt and shame. There were indices that she was quite angry with her father, but that she was afraid that her feelings would be very destructive. Hence, she was self-critical with a severe and harsh super ego. Sonja on the other hand was caught in a struggle of idealizing or devaluing others and herself. She had high expectations that suddenly switched to devaluation, leading her to become disappointed and angry, with low self-esteem.

At one year follow-up, Johanna was outside the clinical range on all measures and reported that she ‘did not have any problems anymore’ (good outcome case). Sonja, on the other hand, at one year follow-up was still in the clinical range in the MADRS and BDI and reported that ‘nothing had improved/was feeling worse’ (poor outcome case).

The therapist for these two patients was a Norwegian male specialist in child and adolescent psychiatry in his 60 s. He was a trained psychoanalytic therapist and had over 30 years of experience in psychodynamic therapy with children and adolescents. In addition, he had attended the one-year FEST-IT training to provide dynamic psychotherapy with or without transference work.

#### 2.1.2. Treatment

Short-term psychoanalytic psychotherapy—STPP [28] was offered including 28 weekly, 45 min sessions. All sessions were audiotaped. The two girls were randomized to STPP without transference work. This was unknown to the authors until after the APQ scorings and the randomization key was opened.

### 2.2. Measures

#### 2.2.1. Diagnostic Instruments

The diagnostic interview M.I.N.I 6.0.0 [29] was used to capture symptom diagnoses and the Structured Interview for DSM-IV Personality (SIDP-IV) [30,31]. Both interviews were completed before, after, and one year after psychotherapy ended.

#### 2.2.2. Outcome Measures

*Psychodynamic functioning scales*—PFS [26]. PFS is based on a psychodynamic interview designed for measuring change, beyond symptoms and general dysfunctions that might take place during and after psychodynamic therapy. As used in FEST-IT it assesses levels of interpersonal aspects (1: quality of family relations and 2: quality of friendships), and intrapersonal aspects (3: tolerance of affects, 4: insight, and 5: problem solving and adaptive capacity) on a scale from 0–100. The PFS have demonstrated good interrater reliability with an Intra Class Correlation (ICC) of 0.82 (CI 0.73–0.90), and a cut off score for reliable change of 6.1 for PFS mean [22].

*Beck Depression Inventory*—BDI-II [32]. The BDI-II is a widely used 21-item self-report inventory measuring the severity of depressive symptoms. The patients filled in the questionnaire before, during, after, and one year after therapy ended. Cut off scores are 0–13 for minimal depression, 14–19 for mild depression, 20–28 for moderate depression, and 29–63: severe depression. The internal consistency of the BDI II has been described as around 0.9 and the retest reliability ranged from 0.73 to 0.96 [33].

*Montgomery Åsberg Depression Rating Scale*—MADRS [34]. MADRS was rated by the therapists during therapy and by independent raters at pre-, post-, and 1 year follow up. In MADRS a score of 0 to 6 represents no symptoms, 7 to 19 mild depression, 20 to 34 moderate depression, and above 34 points are considered severe depression. In the present study, ICC for MADRS single measure was 0.78 (Cl 0.58–0.9).

#### 2.2.3. Process Measures

*Adolescent Psychotherapy Q-Set*—APQ [9,10]. The APQ is a measure developed to describe psychotherapy processes with adolescents in a way that enables quantitative analyses of therapeutic interactions in therapy dyads. The Q-methodology provides a holistic approach for studying phenomena, by exploring how all variables relate to each other, using Q-factor analysis [35]. The unit of analysis is the entire session. In the coding process, 100 items are placed on a scale from (1) ‘extremely uncharacteristic’ to (9) ‘extremely characteristic’. A rating of 5 indicates that the item was irrelevant for the specific session. An item scored as characteristic means that it was saliently present. An item scored as uncharacteristic means that it was conspicuously absent and/or explicitly ‘missing’. Using a forced choice approach the items are placed in a semi-normal distribution. There is a set manual with clear definitions and examples to help in the rating process [9]. Studies have found the APQ to have good reliability and validity [10,36].

#### 2.2.4. Other Measure

*Parental Bonding Instrument*—PBI [37]. The PBI is aimed at measuring perceived characteristics of one’s parents. It is a self-report and measures two parenting styles in both mother and father; that is, the level of ‘care’ and ‘protection’. PBI has been found to have satisfactory reliability and validity [38].

#### 2.2.5. External Ratings and Reliability

All 28 sessions were available from Johanna’s therapy and 24 sessions were available for analysis from Sonja’s therapy (she attended 27 sessions, and three sessions were not audio taped due to technical failure). This sample size is in line with recommendations for Q-factor analysis, as authors have stated that the ratio of participants to variables in this type of analysis should be of 1:2 [39] or that a sample size of 40–50 is enough to provide an adequate picture of the subject under study [40]. Both requisites are met in this study’s sample.

The 52 audio-recorded sessions from the two therapies were coded using the APQ by two master’s level clinical psychology students (2nd and 3rd author) and the last author who had undergone extensive training with one of the developers of the APQ. The sessions were coded in chronological order through a website especially designed for coding APQ [41]. The patients’ treatment outcomes were not revealed to the authors before all sessions were coded.

The material was exported to IBM SPSS version 25 for reliability analysis. Reliability was carefully monitored and 38% of their entire treatments were double coded. Inter-coder reliability for the APQ ratings was measured by intra-class correlations (ICC), using a two-way mixed consistency model. The average reliability of Johanna’s sessions was 0.8 (ranging from 0.79 to 0.92); and the average of Sonja’s sessions was 0.7 (ranging from 0.66 to 0.89), indicating moderate to excellent reliability [42].

### 2.3. Statistical Analyses

After coding, the 52 Q-sorts were merged into one dataset and a Q-factor analysis was performed using the PQMethod software, version 2.35 [43]. Principal Component Analysis was used for factor extraction, because our aim was to find the optimal number of components, the optimal choice of measured variables for each component, and the optimal weights [44]. In other words, instead of finding latent variables the aim was to identify sessions that were similar enough to be considered a group. Varimax was used for factor rotation because it seeks the mathematically best solution, maximizing the amount of variance explained [45]. A five-factor solution was extracted, as this satisfied the Kaiser-Guttman criterion of a minimum eigenvalue of 1.0 [46,47], as well as Brown’s criterion [48] that each factor estimate should be the composite of at least two and preferably three or more statistically significant and non-confounded Q-sorts [45].

The resulting five factors (or interaction structures) accounted for 68.13% of the variance, had a minimum eigenvalue of 6.25, had at least three Q-sorts per factor that were statistically significant at the 0.01 level and were not confounded with another factor. Based on this solution, factor estimates with Z-scores for each APQ item were then computed.

A clinically meaningful name was given to each interaction structure, based on the description of the APQ items with the highest and lowest Z-scores in each factor estimate and a consensus among the investigators. Factor loadings for each factor and patient were plotted for each session, to give a visual representation of the level of each interaction structure during the two therapy trajectories. Factor loadings for the four missing sessions from Sonja’s therapy were averaged from neighbouring loadings. Note that this was done for the visual presentation only; no analysis was performed on these averaged factor loadings.

Microsoft Excel was used for plotting graphs and for calculating average APQ item scores and interaction structure differences. To clarify the differences between the two psychotherapies, average differences on APQ item Z-scores between those interaction structures primarily loaded by Johanna’s therapy, and those primarily loaded by Sonja’s, were computed. These average differences were divided into three groups: those describing therapist actions, those describing the patient’s actions, and those describing features of the interaction between them. With two cases we did not examine statistical significance, just the clinical relevance of these differences.

### 2.4. Ethics

Informed written consent was obtained from all participants before they were included in FEST-IT. FEST-IT was approved by the Regional Committees for Medical and Health Research Ethics (REC) (REK: 2011/1424 FEST-IT). The study was also registered with a ClinicalTrials.gov Identifier: NCT01531101.

## 3. Results

### 3.1. The Two Adolescent Girls at Pre-Treatment

#### 3.1.1. Johanna

Johanna lived together with both of her parents and two younger siblings. Her family had high socioeconomic status. She met the criteria for major depressive disorder, anorexia nervosa, and generalized anxiety disorder based on the M.I.N.I. interview. On SIDP-IV she fulfilled a total 17 personality disorder criteria and fulfilled the criteria for depressive PD (with 5/7 criteria). Her subjective report on depression was in the range of severe depression on the BDI-II, and her therapist rated her in the range of moderate depression on the MADRS (see Table 1). On the PBI, Johanna scored the bond to her mother as ‘high care’ and ‘low overprotection’, indicating an ‘optimal parenting’ bond. She scored the bond to her father as ‘low care’ and ‘high overprotection’, indicating a sense of an ‘affectionless control’ bond.

#### 3.1.2. Sonja

Sonja lived together with her mother and an uncle. Her family had middle to low socioeconomic status. Sonja’s mother had an addiction problem, and their relationship was full of conflicts as her mother seemed to be verbally abusive towards her. She had little to no contact with her father. She had two older brothers who no longer lived at home, but one of them met with her on a regular basis. During her therapy, her uncle became seriously ill. At intake, Sonja met the criteria for panic disorder on M.I.N.I, as well as major depressive disorder. On SIDP-IV she fulfilled a total 13 personality disorder criteria, but no specific disorder. On the PBI, Sonja scored the bond to her mother as ‘low care’ and ‘low overprotection’, indicating ‘neglectful parenting’. She scored the bond to her absent parent as ‘low care’ and ‘high overprotection’, indicating ‘affectionless control’.

### 3.2. Outcome

#### 3.2.1. Level of Psychodynamic Functioning and Diagnostics

The ratings on PFS during treatment and the one-year follow-up period are presented in Table 2.

Johanna was rated as improved on all 5 sub-scales and a reliable change was found from pre- to post treatment and she kept improving to one-year follow-up. In the last interview Johanna’s PFS scores ranged from 71–85. Relationships with family and friends, affect tolerance, insight, and problem-solving capacity were evaluated. Following the PFS manual, this implied that Johanna had: Good stable reciprocally rewarding relationships in her family, conflicts may be painful without compromising basic commitment and security (score of 79); Warm, open, and reciprocally rewarding relationships with friends, other people are generally seen as accepting, trustworthy and responsive (score of 85); Even strong affects are quite well differentiated and flexibly expressed, and symptoms almost never develop (score of 81); She can account for some important inner conflicts, related problems and repetitive behaviour patterns, and personal attitudes, reflects rather freely, but may blame herself or others too much in interpersonal disputes (score of 71); She sometimes curb her own ambitions or is driven towards overachievement. The sense of direction and pursuit of goals are sometimes unclear, but she engages with pleasure in social and recreational activities (score of 80). 

For Sonja, however, ratings on the five PFS sub-scales decreased or stayed the same during therapy. No reliable change on PFS mean was found during the whole study period. Sonja’s PFS scores ranged from 52–70 at follow up, which implies: A tendency to take controlling or submissive roles in the family, were there is limited experience of warmth and trust, avoids conflict or personal pain by keeping distance or self-sacrificing behaviour (score of 52); Relationships with friends might be experienced as problematic by her but may seem normal to others, a tendency to be mildly exploitative, mildly suspicious, dependent or counterdependent in problematic situations (score of 70); Disappointments relatively often lead to denial of affects, outbursts, passive complaining, or symptoms (anxiety, depression, phobias), and less differentiation of feelings (score of 55); Understanding of inner conflicts and associations to past experiences is unclear, or “learned”. Inadequate judgement of herself and others but has ability to observe and reflect with time (score of 60); Develop symptoms or acts inadequately in critical and difficult situations. She might experience restricted pleasure and not dare to initiate desired romantic relationships as well as fails to pursue realistic career goals (score of 55).

Johanna did not longer fulfil any diagnoses, neither on symptom nor personality disorder measures. Sonja on the other hand now fulfilled the criteria for PTSD (an accident happened that terrified her) and social phobia on the M.I.N.I, but she no longer fulfilled the criteria panic disorder. She fulfilled 17 personality disorder criteria and depressive PD (with 6/7 criteria) on SIDP-IV.

#### 3.2.2. Symptoms of Depression

Both patients showed a decrease on BDI and MADRS during the study period (see Table 1).

Johanna was no longer in the clinical range after treatment and at one-year follow up. Sonja’s BDI after therapy was unfortunately missing, but MADRS indicated a score within the range of mild depression, as did both BDI and MADRS at one-year follow-up.

### 3.3. Process 

#### 3.3.1. Interaction Structures

The five interaction structures based on APQ are presented with the help of graphs showing how each structure varies across the whole treatment period of 28 sessions for the two patients. Following the treatment manual, the therapies were divided into an initial phase (the first five sessions), a working phase, and a termination phase (the last five sessions). APQ item numbers will be in parenthesis in the following descriptions.

Interaction structure 1—“Making sense of relationships”

This factor explained the largest proportion of the total variance (26.2% out of a total of 68.1%). In this interaction structure, the young person initiated and elaborated topics (15), described the emotional qualities of the interaction with significant others (6), seemed to be trusting and unsuspicious of the therapist (44), took on board the therapist’s remarks and gave them due consideration (42), felt understood by the therapist (14), discussed and explored interpersonal relationships (63), and went along with the therapist’s attempts to explore thoughts, reactions, or motivations connected to her difficulties (58). The therapist worked with the young person to try to make sense of her experience (9), asked for more information or elaboration (31), and encouraged reflection on internal states and affects (97). As can be seen in Figure 1, Johanna’s sessions almost consistently loaded higher on this factor compared to Sonja’s sessions.

Interaction structure 2—“Working with anger and vulnerability”

Explaining 12.0% of the variance, this factor was the second largest of the five. In this interaction structure, the young person described emotional qualities of the interaction with significant others (6), initiated or elaborated topics (15), expressed angry or aggressive feelings (84), and expressed feelings of vulnerability (8). The therapist made links or salient connections between the young person’s current emotional experience or perception of events with those of the past (92) and drew attention to the young person’s characteristic ways of dealing with emotion (60). In their interaction, it seemed like the young person was calm and composed, even when the therapist was exploring an anxiety-provoking subject or in any other way challenged the young person (10) who also took on board the therapist’s remarks and gave them due consideration (42). It also appeared like the young person felt understood by the therapist (14), as well as showed trust and seemed unsuspicious of the therapist (44).

This factor also differentiates between the two therapies. Except for in the initial phase, Johanna’s sessions had a higher average loading (see Figure 2).

Interaction structure 3—“Fragile self-image”

This factor explained 11.0% of the total variance. In this interaction structure, self-image was a focus of the session (35), and the young person seemed to feel inadequate, inferior (59), shy or self-conscious (61). The therapist worked with the young person to try to make sense of experience (9) and asked for more information or elaboration (31), whilst the young person went along with the therapist’s attempts to explore thoughts, reactions, or motivations connected to her difficulties (58), and handed over the direction of the session to the therapist (87). In this interaction structure, neither the therapist nor the young person focused on their relationship (98). It seems like the young person felt understood by the therapist (14), trusted him, and came forth as unsuspicious of the therapist (44).

This factor on average loaded higher in Johanna’s sessions. As can be seen in Figure 3, this difference was most pronounced during the working phase of the therapies (the middle 18 sessions for Johanna and the middle 14 for Sonja).

Interaction structure 4—“Fearful but suppressed/inhibited”

This factor explained 10.4% of the total variance. In this interaction structure, the therapist tended not to emphasize feelings that the young person found difficult to recognize or accept (50), neither the therapist nor the young person focused on their relationship (98), and there are few silences (12). The young person initiated and elaborated topics (15), discussed and explored current interpersonal relationships (63), described emotional qualities of the interaction with significant others (6), but did not seem curious about the thoughts, feelings, and behaviours of others (23). It also seemed like the young person feared being punished or threatened (16), felt rejected or abandoned (41), and felt unfairly treated (55). 

Sonja’s sessions typically load higher on this factor (Figure 4). This difference in factor loadings was particularly pronounced during the termination phase of the therapy compared to Johanna’s.

Interaction structure 5—“Working with low mentalization”

This last factor explained 8.6% of the total variance. In this interaction structure, the young person did not evidence the capacity to link mental states of self or others with actions or behaviours (24), whilst the therapist worked with the young person to try to make sense of experience (9) and explored interpersonal relationships (63). Furthermore, neither the therapist nor the young person focused on their relationship (98) and talk of interruptions in treatment or endings seemed to be avoided (75). The therapist expressed opinions or took positions either explicitly or by implication (93), asked for more information or elaboration (31), and raised questions about the young person’s view of an experience or event (99). 

On average, in Sonja’s sessions this factor loaded higher than Johanna’s. This is especially pronounced during the working phase but reverses during the termination phase (Figure 5).

#### 3.3.2. Interaction Structure Differences

Table 3 shows the average difference in *the interaction or the dyad’s* nature and atmosphere, Table 4
*the therapist’s* actions and interventions, and Table 5
*the adolescent’s* attitudes, behaviours, and feelings on APQ item Z-scores between those interaction structures primarily loaded by Johanna’s therapy, and those primarily loaded by Sonja’s. A positive difference indicates that this specific APQ item on average was more characteristic in the interaction structures primarily loaded by Johanna’s therapy. A negative difference indicates that it was more evident in Sonja’s therapy.

As can be seen in Table 3, the dyad or *the interaction* between patient and therapist in Johanna’s therapy showed less resistance, control, and rejection by the patient compared to Sonja’s therapy. In addition, Johanna appeared to feel more understood by the therapist and less suspicious of him. Their interaction also demonstrated more shared understanding when referring to events of feelings. In Johanna’s therapy, the therapist also encouraged more reflection, focused more on her characteristic ways of dealing with emotions as well as her emotional states in session, and worked more towards identifying recurrent patterns in her conduct.

Table 4 shows that when *the therapist* was with Sonja, he used more remarks aimed at facilitating her speech, attended more to her somatic feelings and sensations, tended to present events from a different perspective, focused more on her symptoms, made more definite statements about what was going on in her mind, and revealed more of his own emotional responses. With Johanna, on the other hand, the therapist drew more attention to feelings regarded by her as unacceptable, paid more attention to her feelings about interruptions and endings, refrained from taking a position in relation to her thoughts and behaviour, and encouraged her to try new ways of behaving with others and to be more independent. 

As can be seen in Table 5, depicting *the adolescents’* feelings, attitude and behaviour, Sonja showed more irritation and expressed more anger in her sessions than Johanna, she found it more difficult to concentrate or maintain attention in sessions, and tended to blame others or external forces for her difficulties (compared to Johanna). Johanna, on the other hand, was more curious about others and more capable of mentalizing about other people and herself. She also expressed more vulnerability, achieved a greater self-understanding, expressed a greater desire for autonomy, was more self-conscious and expressed more painful affect than Sonja.

## 4. Discussion

The primary aim of this study was to use in-depth methods to elucidate the processes in the psychotherapy of two depressed female adolescents, with the goal of identifying specific processes or interactions associated with poor and good outcome.

The first question of this study focused on what characterized the two girls and the problems they presented before therapy, immediately after, and at one year follow-up. The two patients were 17-year-olds diagnosed with major depression who came from different socioeconomic backgrounds. Both girls had some difficulties relating to their parents, but the problems Sonja faced were more severe, even if the different measures employed did not capture this at pre-treatment. At the start of treatment both reported moderate problems in daily life functioning. Symptom levels were quite high, and they had comorbid diagnoses (such as panic disorder and eating disorders) which seems common in regular public outpatient clinics.

The patients had different outcomes and the PFS indicated that when therapy ended there was ‘good outcome’ in Johanna’s case, and ‘poor outcome’ in Sonja’s case. PFS is a complex measure that includes more than symptom reduction, as it aims to measure general adaptive functioning and life satisfaction [26]. Measuring both interpersonal and intrapersonal aspects, PFS is a unique measure to capture psychological functioning related to several factors in an adolescent’s life [22]. If only symptoms of depression were used as outcome measure, Sonja would also be classified as a ‘good outcome case’ with large changes on both BDI and MADRS. Hence, to capture clinically significant outcomes, the importance of including other measures than mere symptoms, is accentuated.

To answer the second question of this study, we examined the interaction structures between the therapist and patients using the APQ. Five interaction structures were identified: (1). Making sense of relationships, (2). Working with anger and vulnerability, (3). Fragile self-image, (4). Fearful but suppressed/inhibited, and (5). Working with low mentalization. The first three interaction structures could be conceived as subdivisions of the first interaction structure found in Calderon and colleagues [13] and in Fredum and colleagues [17], i.e., reflecting a pattern in psychodynamic therapy in which the adolescent client is willing to cooperate and explore difficult feelings and experiences. Similarly, the fourth and fifth interaction structures of this study could be argued correspond to the third pattern found in Calderon and colleagues [13] and the second pattern in Fredum and colleagues [17] displaying a patient who does not show capacity for mentalization and cannot express vulnerability. In the present study the patient seemed somewhat more engaged in the therapeutic endeavour.

As a third question of this study, we asked whether the interaction structures differed in the two treatments over time. The analysis showed that they could roughly be divided between the two therapies. Out of the five interaction structures extracted, three (Making sense of relationships; Working with anger and vulnerability and Fragile self-image) were particularly representative of Johanna’s therapy, whilst the remaining two (Fearful but suppressed and Working with low mentalization) represented mainly the patient who did not profit from time-limited psychodynamic therapy (Sonja).

The interaction structures found suggest that Johanna’s treatment was based on cooperation and trust in the therapist, as well as displaying a therapist working within the psychodynamic treatment model presented in general terms by Shedler [49] and in the specific treatment manual [28], with a patient who partakes in the therapist’s invitations to work therapeutically. The findings further suggested that there was a focus on interpersonal relationships throughout most of Johanna’s therapy. Moreover, the patterns left the impression that while the themes within the sessions varied, from more to less challenging areas, the therapeutic bond remained strong. Through the initiation phase and the first half of the working phase, the interaction structures *Making sense of relationships* and *Fragile self-image* were particularly representative of the interaction between Johanna and the therapist, giving an impression of Johanna as willing to remain open to therapy, even when this made her feel shameful and self-conscious. Towards the end of therapy, Johanna seemed to be more in touch with anger and vulnerability, and the therapist appeared to explicitly work with her emotions and drew connections to how her current emotional experiences could be linked to those of the past. This shift in focus towards the end can be explained by external events making anger and vulnerability more accessible. However, it could also well be an expected result of constructive work on making these emotions more acceptable for the patient.

One the other hand, the global description of Sonja’s therapy was that she from the beginning of therapy showed limited capacity for mentalization, and the therapist seemed to become increasingly active in both expressing his personal opinions as well as exploring Sonja’s views and experiences. In the mid working phase of the therapy, the interaction structures indicated that the alliance seemed to have evolved, and that the therapy relationship was more cooperative, as her sessions in this phase was mainly represented by the *Making sense of relationships* factor. Towards the end of therapy, the patient focused on difficult relationships in which she feared being mistreated or felt rejected. The therapist seemed to neglect feelings that the patient found difficult to accept, and the patient did not seem curious about the thoughts and feelings of others. The APQ items concerning transference work was seen as negatively salient in factor four and five, underscoring the lack of working in the here and now with the ongoing relationship. The trajectory of Sonja’s therapy seemed to be highly impacted by her difficult living situation, where conflicts with her parent occurred frequently. Life events seemed to attenuate her therapeutic change, a finding that is in line with prior research underlining the inherent challenges related to the stress of living at home that young people may experience [50]. Although both adolescents lived at home, Johanna’s family environment was quite different from Sonja’s. Johanna did have a complicated relationship with one of her parents but had the support of the other parent. Sonja did not have much support, she lived with only one parent who had problems with addition and was verbally abusive towards her, while the other parent was absent. These differences probably impacted how the two girls related to the therapy work and to the relationship with the therapist.

The fourth and final research question explored whether there were observable differences in patient behaviour, therapist behaviour, and/or relational aspects in the therapies. Our findings indicated that there were notable differences with regard to the young person’s ability to mentalize herself and others, making it more difficult for Sonja to understand herself in relation to others compared to Johanna. In turn, this would likely represent an impediment to the identification and working with central emotional and relational themes associated with her difficulties. 

A person’s capacity for mentalization is developed through his or her attachment to primary caregivers [51,52]. On the PBI, Sonja rated the bond to the parent with whom she lived as indicating ‘neglectful parenting’. Sonja’s relationship to the parent she lived with was described in her sessions as being deeply ambivalent; she could at one time express profound anger and hatred towards the parent, and at another express warmth and sympathy. A similar ambivalence was apparent in her interactions with the therapist. As mentioned, Sonja seemed to express less vulnerability and painful affect than Johanna, which suggests that she was less comfortable with intimacy, a central aspect of secure attachment [53]. Sonja’s manner of relating to her parent and to the therapist may be a consequence of her having developed an anxious-ambivalent attachment style, and that this attachment style might both have been a factor in the development of her difficulties as well as impeding the development of an emotional bond with her therapist, and ultimately the outcome of her therapy.

In relation to therapist’s behaviour, the most characteristic APQ items indicated that in Johanna’s therapy the therapist actively encouraged her to explore and verbalize her thoughts and feelings, he focused on Johanna’s feelings about what happened or was said in the sessions, he drew Johanna’s attention to how she usually dealt with emotions, he pointed out recurrent patterns in Johanna’s behaviour, and emphasized feelings considered by Johanna as inappropriate, wrong, or dangerous. These are all features that are considered prototypical of psychodynamic treatment [49]. As these items were rated less characteristic in the interaction structures primarily expressed in Sonja’s therapy, the therapist used more prototypically psychodynamic interventions in Johanna’s therapy compared to in Sonja’s therapy. It seems like the therapist’s psychodynamic approach was modified in his work with Sonja, as the differences in interaction structures indicated that he more often restated Sonja’s descriptions and urged her to look at situations differently, he encouraged more reflection on Sonja’s symptoms, he appeared to spend more time linking sessions together, and he revealed more emotional responses, perhaps trying to forge a stronger and more intimate bond with her. So far, we do not know if this modification represented a constructive way of dealing with a weaker alliance and more resistance from the patient.

In summary, the interaction structures presented in these two cases indicated that symptom reduction and psychological change happen in the context of a good working relationship. The ability to mentalize self and others and be psychologically minded differed between the two patients, leading to a different level of receptivity in the two patients. This is in line with prior research indicating that to some degree the competence of the therapist (i.e., in delivering therapeutic interventions according to the underlying theory, in a responsive and well-timed manner), is a function of how receptive a client is [54]. In our study, the same therapist treated both patients—which supports such a conclusion. The interaction structures imply that both Sonja and Johanna got to a point where they were doing therapeutic work, but that it took Sonja longer to get there, and that the process stagnated towards the end. It may be that for Sonja, becoming more psychologically aware through the process, also made her more aware of her psychological pain. Her limited capacity to experience and express her own emotions can be understood as a defence mechanism developed to protect her from the psychological hardships in her life. Despite this, we should also acknowledge that the treatment was relatively short-term; some patients may require more time to gain improvement. Through a research literature review, De Geest & Meganck [55], found that the time limit of sessions is anything but a neutral intervention; it is a technique that complexly interacts with therapy processes on multiple levels. When examining the interaction structures, it might have been that the time limit of 28 sessions was not helpful for Sonja. In addition, Sonja was still living at home in a dysfunctional relationship with a drug abusing parent, preventing her from making optimal use of the treatment provided. 

In addition to answering our research questions, we propose further reflections regarding the two cases and its implications. We know from previous research in the field of personal recovery that we need to take the clients’ subjective account into consideration when establishing the outcome of treatments [56]. Although both patients reported reduced depressive symptoms at follow-up, Sonja’s subjective experience one year after treatment was that nothing had improved or that she might even be feeling worse. In sharp contrast, Johanna reported having no problems at all. The fact that improvement after STPP consists of more than symptom reduction, is in accordance with Busch and colleagues [57] who underline that if psychodynamic treatment is successful, patients will be better at managing depressed feelings and aggression, less prone to guilt and self-devaluation, make more realistic assessment of their own and others’ behaviour and motivations, have more agency, and a more realistic view of their responsibilities and the difference between fantasy and reality, as well as less vulnerable to depression in the face of loss, disappointment and criticism. Johanna’s central depression dynamics was considered at pre-treatment to revolve around guilt and shame, she was angry with her father and afraid of being too destructive, she was self-critical with a severe and harsh super ego. The development of interaction structures throughout the treatment suggested that the therapist emphasised and worked with her central depression dynamics, which may have led to the increased psychological capacities as observed with PFS connecting process to outcome. Sonja was still caught in a struggle of idealizing or devaluing others and herself, sustaining her low self-esteem. 

Both girls received STPP without transference work. Even if Sonja did not fulfil the criteria for a personality disorder using the SIDP-IV, she experienced relational trauma from a young age and presented varied and shifting symptom diagnoses. It has been argued that women with a lifelong pattern of poor object relations might especially profit from transference work to respond well to STPP [58]. In line with this, it seems that adolescents with cluster B personality disorder symptoms also do better with transference work than without [59]. The APQ interaction structure *Working with low mentalization* indicated that Sonja struggled with mentalising herself and others. Transference work could have given Sonja a chance to reflect on her own mind and her ideas about what went on in the therapists mind in the here and now, which might have moved the therapy forward. 

So far, five different studies using the APQ have indicated that when there are difficulties in the therapy, the therapists deviate from their mode of treatment. Most authors argue that this is a flexible strategy, and a way to accommodate and hopefully make the patient respond to new interventions [13,60]. Yet, one could also ask if those deviations reflect countertransference responses on the part of the therapist who more or less unconsciously seeks to avoid emotional hardship or whether it is thoughtful intended deviations from the treatment model. In any case, further research is needed to find ways to help those adolescents that do not engage in the therapy nor initially respond well to theory specific techniques and interventions in the treatment of adolescent depression. 

## 5. Strengths and Limitations 

Coding whole therapies has the advantage that it creates a complete ‘motion picture’ of what happens in therapy with the two adolescents, without having to extrapolate based on a limited number of sessions. Another strength was that the two adolescents were treated in a standard outpatient setting. Thus, their problems, symptoms and therapies might be relevant and generalizable to real life treatments in an outpatient setting.

Notwithstanding this, there are several limitations in this study that need mentioning. When selecting the participants, the aim was to find two patients who were as similar as possible who were treated by the same therapist but with different outcomes. Sonja and Johanna shared many characteristics, however, when listening to the sessions it was apparent that there were substantial differences between the two at the outset, i.e., regarding their families’ socioeconomic status and conflict level, as well as their level of mentalization, as mentioned above. This can be regarded as a limitation when comparing the two, however, it also reflects the reality of clinical practice, and emphasizes the fact that most patients do not match the criteria of being an ‘average patient’.

Another limitation of this study was that there was no information regarding the level of representativeness of this therapist’s responses or techniques in relation to a typical psychodynamic therapist, as the APQ’s STPP prototypes were not developed by the time this research was conducted. However, the most important aspect of this research was to explore in-depth the interactions of the therapist and clients and ensure transparency. One strength of case studies like the present one, is their ability to examine variation within a phenomenon or a process, not necessarily within a population, and generate rich descriptions of process data in order to investigate potential associations between process and outcome and mechanisms of change from which to build new theories or nuance existing ones [23,24,61]. The two cases were chosen to investigate the processes associated with divergent outcomes. However, one may question the characterization of Sonja’s outcome as poor. Her symptom scores showed a reduction that also could have been characterized as a positive outcome. It might be that Sonja’s therapy made her more capable of experiencing and accepting emotions, making her difficulties more apparent and present to her, and hence, easier for her to communicate.

Overall inter-rater reliability was higher in Johanna’s sessions than in Sonja’s sessions. Post-reliability checking discussions suggested that the raters did not disagree on the understanding of the session, but on how to code Sonja’s responses. In many of Sonja’s sessions she could in parts express vulnerability and in other parts distance herself from vulnerability. This ambiguity in the sessions made it difficult for the raters to decide if it was characteristic or uncharacteristic. This kind of ambiguity or ambivalence is something that clinically cannot be avoided, especially with this group of patients. However, it does represent a limitation with regard to the reliability of the coding and possibly a limitation of the APQ approach as well.

Finally, it is important to note that other similar studies have coded sessions randomly. Raters’ expectations regarding the process and outcome (which was revealed once all the sessions had been rated) might have been influenced their coding. However, the general impression was that this did not affect the rating of the sessions, as reliability was satisfactory and did not improve throughout the process.

## 6. Conclusions

Using the Adolescent Psychotherapy Q-set method, the current study examined and contrasted interaction structures between one therapist and two depressed 17-year-old girls, Johanna (good outcome) and Sonja (poor outcome), in short-term psychoanalytic therapies selected from an RCT study. The method meaningfully distinguished between the two processes, showing that the therapist used more prototypically psychodynamic interventions in the good outcome versus the poor outcome case and that the patient in the latter might have been less receptive to the therapists’ interventions, which in turn seemed to have been linked to her facing a far more emotionally challenging and less supportive life situation. Our findings shed further light on the intricate dynamics of the therapist-patient dyad, in which the therapist’s responsiveness and patient’s receptiveness are crucial but not necessarily stable entities.

## Figures and Tables

**Figure 1 ijerph-19-16939-f001:**
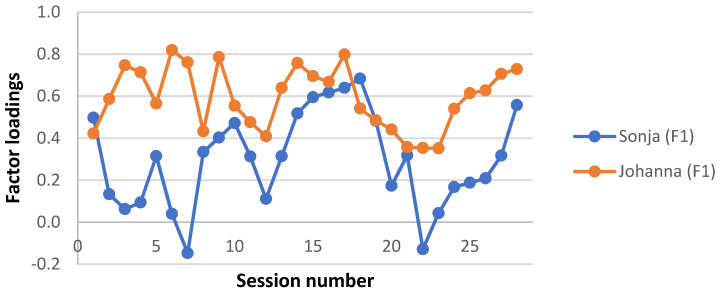
Interaction structure 1: “Making sense of relationships”. Factor loadings on each session for Sonja and Johanna.

**Figure 2 ijerph-19-16939-f002:**
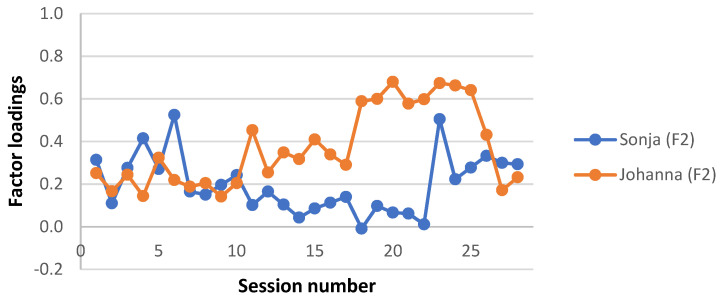
Interaction structure 2: “Working with anger and vulnerability”. Factor loadings on each session for Sonja and Johanna.

**Figure 3 ijerph-19-16939-f003:**
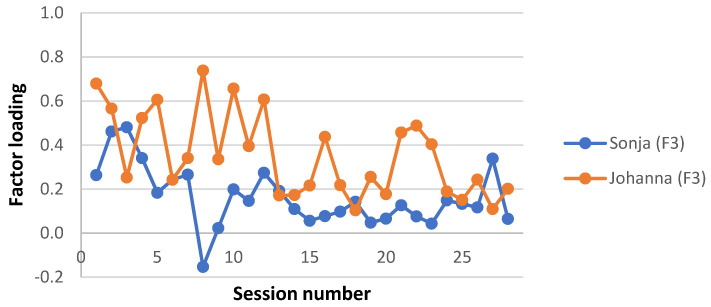
Interaction structure: “Fragile self-image”. Factor loadings on each session for Sonja and Johanna.

**Figure 4 ijerph-19-16939-f004:**
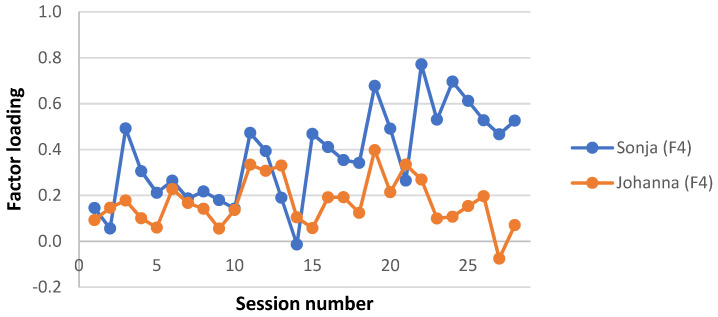
Interaction structure 4: “Fearful but suppressed”. Factor loadings on each session for Sonja and Johanna.

**Figure 5 ijerph-19-16939-f005:**
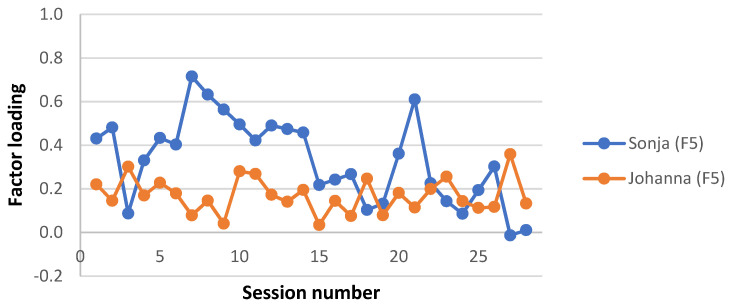
Interaction structure 5; “Working with low mentalization”. Factor loadings on each session for Sonja and Johanna.

**Table 1 ijerph-19-16939-t001:** MADRS and BDI scores for Johanna and Sonja.

	Johanna		Sonja	
	MADRS	BDI	MADRS	BDI
Pre-treatment	32	37	20	35
Session 12	10	26	25	31
Session 20	3	13	12	12
After therapy	8	5	19	-
One-year follow-up	2	4	8	18

Cut off scores MADRS: 0 to 6—no symptoms, 7 to 19—mild depression, 20 to 34—moderate depression, and >34—severe depression. Cut off scores BDI II: 0 to13—minimal depression, 14 to 19—mild depression, 20 to 28—moderate depression, and 29 to 63—severe depression.

**Table 2 ijerph-19-16939-t002:** PFS scores for Johanna and Sonja at three time points.

	Pre-Treatment		Post-Treatment		One-Year Follow-Up	
	Johanna	Sonja	Johanna	Johanna	Sonja	Johanna
Family	66	60	72	52	79	52
Friends	77	76	74	70	85	69
Tolerance for affect	64	55	75	55	81	59
Insight	68	60	74	61	71	61
Problem solving	63	57	74	57	80	54
Mean	68	62	74	59	79	59

**Table 3 ijerph-19-16939-t003:** Interaction structure differences measured in Z-scores on APQ Items describing the interaction between the young person (YP) and therapist (T).

Item	Description	Difference
58	YP resists T’s attempts to explore thoughts, reactions, or motivations related to problems	−2.52
87	YP is controlling of the interaction with T	−2.07
42	YP rejects T’s comments and observations	−1.68
60	T draws attention to YP’s characteristic ways of dealing with emotion	1.56
14	YP does not feel understood by T	−1.52
97	T encourages reflection on internal states and affects	1.42
96	T attends to the YP’s current emotional states	1.40
12	Silences occur during the session	0.81
62	T identifies a recurrent pattern in YP’s behavior or conduct	0.69
5	YP has difficulty understanding T’s comments	−0.66
38	T and YP demonstrate a shared understanding when referring to events or feelings	0.59
98	The therapy relationship is a focus of discussion	0.58
56	Material from a prior session is discussed	−0.55
44	YP feels wary or suspicious of the T	−0.54

A positive difference indicates that this APQ item on average was more characteristic of Johanna’s therapy, a negative difference indicates that the item was more characteristic of Sonja’s.

**Table 4 ijerph-19-16939-t004:** Interaction structure differences measured in Z-scores on APQ items describing therapist (T) actions.

Item	Description	Difference
3	T’s remarks are aimed at facilitating YP’s speech	−1.44
50	T draws attention to feelings regarded by YP as unacceptable	1.33
75	T pays attention to YP’s feelings about breaks, interruptions, or endings in therapy	1.30
93	T refrains from taking position in relation to YP’s thoughts or behavior	1.05
77	T encourages YP to attend to somatic feelings or sensations	−0.94
85	T encourages YP to try new ways of behaving with others	0.73
80	T presents an experience or event from a different perspective	−0.68
39	T encourages YP to reflect on symptoms	−0.62
48	T encourages independence in the YP	0.54
89	T makes definite statements about what is going on in the YP’s mind	−0.54
81	T reveals emotional responses	−0.52

A positive difference indicates that this APQ item on average was more characteristic of Johanna’s therapy, a negative difference indicates that was more characteristic of Sonja’s.

**Table 5 ijerph-19-16939-t005:** Interaction structure differences measured in Z-scores on APQ items describing the young person’s (YP) actions and experiences.

Item	Description	Difference
10	YP displays feelings of irritability	−2.25
24	YP demonstrates capacity to link mental states with action or behavior	1.91
8	YP expresses feelings of vulnerability	1.90
23	YP is curious about the thoughts, feelings, or behavior of others	1.35
32	YP achieves a new understanding	1.27
29	YP talks about wanting to be separate or autonomous from others	1.19
84	YP expresses angry or aggressive feelings	−1.06
67	YP finds it difficult to concentrate or maintain attention during the session	−1.00
34	YP blames others or external forces for difficulties	−0.95
61	YP feels shy or self-conscious	0.93
26	YP experiences or expresses troublesome (painful) affect	0.93

A positive difference indicates that this APQ item on average was more characteristic of Johanna’s therapy, a negative difference indicates that was more characteristic of Sonja’s.

## Data Availability

The data is available from the last author.

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
