# Peer review of "Examining Psychotherapeutic Processes with Depressed Adolescents: A Comparative Study of Two Psychodynamic Therapies"

_ijerph, 2022, doi:10.3390/ijerph192416939_

Round 1
Reviewer 1 Report
The primary aim of this study was to use in-depth methods to elucidate the processes In the psychotherapy of two depressed female adolescents, with the goal of identifying specific processes or interactions associated with poor and good outcome. The objective of the study was accomplished with clarity and scientific rigour. As stated by the authors, given the sampling difficulties, it would be useful to broaden the sample to which to submit the treatment for future studies.
Author Response
Dear Reviewer 1,
It is our understanding that there was nothing to address from this reviewer for the present manuscript.

Reviewer 2 Report
Questions about methodology:
1. In Q-factoring, the procedure involves the classification of people and require a certain number of participants in relation to the number of items (e.g., items on the APQ). Is your methodology veridical Q factor analysis?
2. Why did you use principal components extraction rather that factor analysis (e.g., centroid)
3. Why did you use varimax rotation rather than an oblique rotation?
Author Response
Reviewer #2
- In Q-factoring, the procedure involves the classification of people and require a certain number of participants in relation to the number of items (e.g., items on the APQ). Is your methodology veridical Q factor analysis?
Authors’ response: We believe the methodology is veridical Q-factor analysis. In response, we have added the following sentences to the methods section:
“This sample size goes in line with recommendations for Q-factor analysis, as authors have stated that the ratio of participants to variables in this type of analysis should be of 1:2 [38] or that a sample size of 40-50 is enough to provide an adequate picture of the subject under study [39]. Both requisites are met in this study’s sample.”
- Why did you use principal component extraction rather than factor analysis (e.g., centroid)?
Authors’ response: We added the following sentence to the methods section:
“Principal Component Analysis was used for factor extraction, because our aim was to find the optimal number of components, the optimal choice of measured variables for each component, and the optimal weights [43]. In other words, instead of finding latent variables the aim was to identify sessions that were similar enough to be considered a group.”
- Why did you use varimax rotation rather than oblique rotation?
Authors’ response: As rotation does not change the results but only makes them easier to interpret, we chose varimax rotation to maximise the amount of variance explained. We added the following to the methods section:
“Varimax was used for factor rotation because it seeks the mathematically best solution, maximizing the amount of variance explained [44].”
